# Healthcare system action on employment as a social determinant of health in people living with HIV: A qualitative study

Amy Craig-Neil[1,2], Julia Ho[1,2], Melissa Perri[1,2,3], Mark Gaspar[3], Charlotte Hunter[4,5], Beth Rachlis[3,6], Claire E. Kendall [6,7,8,9], Sergio Rueda[10,11], Ann N. Burchell[2,3,4,5], Andrew D. Pinto [1,2,3,4,5,12] *

1 Li Ka Shing Knowledge Institute, Upstream Lab, MAP/Centre for Urban Health Solutions, Unity Health Toronto, Toronto, Canada, 2 Li Ka Shing Knowledge Institute, MAP Centre for Urban Health Solutions, Unity Health Toronto, Toronto, Canada, 3 Dalla Lana School of Public Health, University of Toronto, Toronto, Canada, 4 Faculty of Medicine, Department of Family and Community Medicine, University of Toronto, Toronto, Canada, 5 Department of Family and Community Medicine, St. Michael's Hospital, Toronto, Canada, 6 ICES, Toronto, Canada, 7 Department of Family Medicine, University of Ottawa, Ottawa, Canada, 8 Bruyère Research Institute, Ottawa, Canada, 9 Institut du Savoir Montfort, Montfort Hospital, Ottawa, Canada, 10 Institute for Mental Health Policy Research and Campbell Family Mental Health Research Institute, Centre for Addiction and Mental Health, Toronto, Canada, 11 Department of Psychiatry, Faculty of Medicine, University of Toronto, Toronto, Canada, 12 University of Toronto Practice-Based Research Network, Toronto, Canada

* andrew.pinto@utoronto.ca

**Data Availability Statement:** All relevant data are within the manuscript.

## Abstract

### Background

Employment is a key social determinant of health. People living with HIV (PLWH) have higher unemployment rates than the general population. Vocational rehabilitation services have been shown to have significant and positive impact on employment status for PLWH. Understanding whether integrating vocational rehabilitation with health care services is acceptable, from the perspectives of PLWH and their health care providers, is an area that is understudied.

### Methods

We conducted a qualitative study and collected data from focus groups and interviews to understand the perspectives of stakeholders regarding the potential for vocational rehabilitation and health care integration. We completed five focus groups with 45 health care providers and one-to-one interviews with 23 PLWHs. Participants were sampled from infectious disease, primary care clinics, and AIDS Service Organizations in Toronto and Ottawa, Canada. Interviews were audio-recorded and transcribed. We conducted a reflexive thematic analysis of the transcripts.

### Findings

We found health care providers have little experience assisting patients with employment and PLWH had little experience receiving employment interventions from their health care

**Funding:** This work was supported by the Canadian Institutes of Health Research (www.cihr-irsc.gc.ca) through a Catalyst Grant (201710CDP), with ADP as Principal Investigator. AP is supported as a Clinician Scientist by the Department of Family and Community Medicine, Faculty of Medicine, University of Toronto, the Department of Family and Community Medicine, St. Michael's Hospital, and the Li Ka Shing Knowledge Institute, St. Michael's Hospital. AP is also supported by a fellowship from the Physicians' Services Incorporated Foundation, as the Associate Director for Clinical Research at the University of Toronto Practice-Based Research Network (UTOPIAN), and as a Canadian Institutes of Health Research Applied Public Health Chair. Dr. Claire Kendall is supported by a CIHR-Ontario HIV Treatment Network New Investigator Award in HIV/AIDS Health Services/ Population Health Research. The funders had no role in study design, data collection and analysis, decision to publish, or preparation of the manuscript.

**Competing interests:** The authors have declared that no competing interests exist.

team. This lack of integration between health care and vocational services was related to uncertainties around drug coverage, physician role and living with an episodic disability. Health care providers thought that there is potential for a larger role for health care clinics in providing employment interventions for PLWH however patients were divided. Some PLWH suggest that health care providers could provide advice on the disclosure of status, work limitations and act as advocates with employers.

## Interpretation

Health care providers and some PLWH recognize the importance of integrating health services with vocational services but both groups have little experience with implementing these types of interventions. Thus, there needs to be more study of such interventions, including the processes entailed and outcomes they aim to achieve.

## Introduction

Employment is a key social determinant of health [1–3]. Employment determines one's income, and therefore one's ability to afford basic necessities including food, stable housing, clothing and medications [4,5]. Employment is also central to our relationships and connections to others: how others see us, the identity that we project to others, and how we build social capital [6–9].

Studies have found that unemployment rates in people living with HIV (PLWH), estimated to be between 45–65%, are higher than the 5–10% unemployment rate in the general population [10–14]. Underemployment, precarious employment and unemployment conditions for PLWH are associated with poor health outcomes and poor quality of life [15–17].

There are many factors that contribute to the decision for PLWH to seek employment including a desire to make a productive contribution to society, to feel normal, and to increase personal income [18,19]. In addition, PLWH may face significant pressure from family, friends and health providers to engage in paid work [20,21]. However there are many barriers including uncertainty around losing disability benefits, particularly coverage for expensive antiretroviral therapy, and a fear of being trapped in a job to then maintain drug benefits [19,22–24]. These concerns can vary by country and by state or province. Some PLWH fear that work could worsen their health status [21,24] and that their need to take off time from employment to attend to their health (e.g., attend appointments and tests, take short-term disability) may not be accommodated [25,26] or that their status may be disclosed in the workplace. PLWH have significant experience with and fear of workplace discrimination based on their HIV status [23,24,26,27]. These barriers typically have a greater impact among PLWH with worse health status and among those who have been away from the workforce longer [26].

Vocational rehabilitation (VR) services have been shown to improve employment outcomes for PLWH [28]. VR programs provide training and job placement, and also assistance with finances and debt, money management, housing access, access to health care, addressing mental health and addictions, addressing outstanding legal issues and building social relationships [29]. An analysis of 104 participants of a workforce re-entry demonstration project in California found that the most commonly used resources were retraining and mental health resources [30]. In another study, recipients reported VR increased their self-confidence and self-respect, provided motivation to continue to search for jobs, and facilitated adjustment when entering the workforce [31]. A study of 53 PLWH who took part in a group VR

intervention in New York City that focused on goal setting and developing strategies for managing health, work and daily life stressors resulted in moderate to large improvement in the perception of ability to balance work, health and daily life [32]. These VR programs are typically delivered by traditional employment agencies and advisors but can be linked with health services through referrals. While they have shown to be successful, research also indicates that uptake by PLWH into these employment programs is low [22,33]. Some reasons for low uptake include lack of knowledge of VR, self-confidence in maintaining a job, general health perception [33] and health insurance coverage [34].

Few studies have examined the integration of VR services in health care settings. Small studies in the UK have found such efforts to be successful for other conditions. For example, general practices in eastern England that care for patients with mental illness have incorporated employment advisers to support people hoping to gain employment or improve their existing employment conditions [35]. Another pilot project found that Employment Advisers from a social service agency located in GPs' surgeries was rated highly by participants [36]. One study reported that these interventions reach individuals who would otherwise not access vocational services, are proactive and anticipate challenges, and focus upon client participation [37]. There are even fewer that have a focus on PLWH. A systematic review of employment interventions in health settings found only one study focused on PLWH [38].

Given the paucity of information on employment interventions for PLWH in health settings and the encouraging results of studies that have integrated employment within health care for other conditions, our objective was to examine the views of health care providers and PLWH about the current and potential role that integrating VR services into health care setting could play in addressing employment and to explore the feasibility of such an intervention.

## Methods

### Design

We conducted a qualitative study to understand the perspectives of health care providers and PLWHs. The study team consisted of clinicians with a focus in HIV (CH, CK, ADP), a qualitative researcher (MG), researchers in HIV care (BR, CK, ANB, ADP, SR) and research staff and students with experience in qualitative methodology (ACN, MP, JH). Our team consulted with representatives from Employment ACTion, a VR service for PLWH in Toronto. The theoretical orientation of this work arose from the recognition of the importance of employment as a key social determinant of health [1–4] and the understanding that PLWH are more likely to be unemployed than the general population [10–14].

### Setting

The study was conducted in two Canadian cities in Ontario: Toronto, the city with the highest incidence of HIV in Canada [39], and Ottawa as a comparable urban centre with the second highest incidence of HIV in the province of Ontario [39]. The study was approved by the Unity Health Toronto, Bruyère Continuing Care and Ottawa Health Science Network Research Ethics Boards.

### Sampling and data collection

We conducted in-depth qualitative focus groups and one-to-one interviews with health care providers. We used purposive sampling to recruit health care providers who worked at primary care and specialty HIV clinics. We included any health care providers who were providing care for patients living with HIV. Participants were recruited by email from local site

contacts. All focus groups were conducted in person at the clinic sites. When in person focus groups were not possible due to COVID-19 restrictions, we conducted individual interviews with providers virtually. The focus groups and interviews were conducted using a semi-structured focus group guide (S1 Appendix) by a trained female research coordinator with an MSc (ACN) or the male Principal Investigator with an MD and MSc (ADP) who both have significant experience in qualitative research. Both focus group facilitators work at the Upstream Lab whose research focus on the social determinants of health (SDOH). They both have an interest in HIV and how it intersects with SDOH.

We also conducted interviews with PLWHs who were unemployed but actively seeking employment and between the ages of 18–64. Participants were recruited using advertisements (posters and emails) at infectious disease and primary care clinics and AIDS Service Organizations in both Toronto and Ottawa. The interviews were completed in person or over the phone. All the interviews were conducted by a research coordinator (ACN) using a semi-structured interview guide (S2 Appendix). Unlike the participants, the facilitator was employed and had been for 2.5 years and not living with HIV at the time of the interviews. She is white and was born in Canada. Data from these interviews has been presented in greater depth elsewhere [40].

Prior to starting the focus groups and interviews, the facilitator introduced themselves and the study team and explained the reasons for doing the research. Written informed consent was obtained for in-person interviews and focus groups. Verbal informed consent was obtained using a verbal consent checklist completed by the facilitator for virtual interviews.

Both the focus group guide and interview guides were developed by a team of stakeholders including clinicians and researchers focusing on HIV, representatives from AIDS service organizations and PLWH. The focus groups explored provider awareness of employment as a social determinant of health of PLWH and available community resources, how they approach PLWH about employment and what could support them in this conversation. The interviews explored the participants' personal experiences with work, unemployment and re-entry and a discussion of how their health care team has or could be involved. We conducted interviews and focus groups concurrently with interim analysis informing and adapting the interview guide.

## Analysis

The interviews and focus groups were audio-recorded and transcribed by a professional transcription company. Field notes were not used. We completed interviews and focus groups until no new ideas emerged from our concurrent analysis, at which point we determined that the data had reached theoretical saturation [41]. No repeat interviews or focus groups were done and transcripts were not returned to participants for comment.

We conducted a reflexive thematic analysis of the transcripts using Braun and Clarke's approach [42]. A subset of the study team members (ACN, MP, MG, CH and ADP) independently read one transcript from a provider focus group and one transcript from an interview with a PLWH to determine emerging codes. Each member then met to discuss and agree on an initial coding framework. The remaining transcripts were read and coded by two members of the study team (ACN and MP) using NVivo 11. The coding framework was expanded and refined iteratively. Once coding was complete, all the codes were extracted and collated. Codes were examined and initial themes were generated from the coded data. These themes were reviewed and refined through examination of all the codes and transcripts. The final themes were determined by consensus by the entire study team. Participants did not provide feedback on the findings.

## Results

Four focus groups and one interview were conducted in Toronto and Ottawa with health care providers. They took place between May 2019 and October 2020 and lasted between 39 and 49 minutes. The four focus groups took place at the clinic location and no one else was present besides participants and researchers. The one interview took place virtually over zoom. There were a total of 45 provider participants (Table 1). Focus groups were advertised internally therefore we do not know how many eligible participants were aware of the study and chose not to participate. No participants dropped out.

Twenty-three PLWH who were unemployed but interested in reentering the workforce were interviewed (17 in Toronto and 6 in Ottawa). The interviews took place between March 2019 and March 2020 and lasted between 15 and 75 minutes. The interviews in Toronto took place in an office space with no one else present besides the participant and the researcher. The interviews in Ottawa took place over the phone. No repeat interviews were completed. The majority of participants were between the ages of 40–55 (n = 12), with seven between the ages of 26–39 and four between the ages of 56–64 with a median age of 48 and standard deviation of 8.5. There were sixteen men and seven women who participated. Most participants were native English speakers (n = 15), but only about half (n = 11) were born in Canada. There were varying levels of education, but most had completed University or College (n = 11) or some University or College courses (n = 5). The study was advertised widely, therefore we do not know the exact number of participants who saw the advertisement but who declined participation. There were people who contacted the study team who did not end up participating in an interview: 11 people were ineligible and 14 people chose not to participate. No participants dropped out or withdrew their data.

### Current role of health care providers in assisting with employment

We found that health care providers and patients have little experience with employment interventions in health settings beyond making or receiving referrals to community resources and providing or receiving letters for workplace accommodations to employers. We identified three common themes to explain this limited experience including uncertainty around drug coverage and Ontario Disability Support Program (ODSP), uncertainty around the role of providers in assisting patients with employment and uncertainty with the ability to work due to the episodic nature of HIV.

**Table 1. Health care providers demographic information.**

|  | ID clinics | Primary Care clinics |
|---|---|---|
| Location | Toronto and Ottawa | Toronto |
| Number of participants | 25 | 20 |
| Number of focus groups or interviews | 3 | 2 |
| Professions | Physician, Registered Nurse, Social Worker, Researcher, Pharmacist | Physician, Registered Nurse, Social Worker, Nurse Practitioner, community researcher |
| Gender<br>　Male<br>　Female | <br>7 (28%)<br>18 (72%) | <br>11 (55%)<br>9 45%) |
| Age Range<br>　18–25<br>　26–39<br>　40–55<br>　56–64<br>　65+ | <br>4 (16%)<br>5 (20%)<br>11 (44%)<br>5 (20%)<br>0 (0%) | <br>2 (10%)<br>10 (50%)<br>6 (30%)<br>1 (5%)<br>1 (5%) |

**Uncertainty around drug coverage and ODSP.** A main concern among providers was continued drug coverage for patients who were considering going back to work. They noted that many of their patients shared their concerns. Many PLWH in Ontario are on ODSP which provides drug coverage, extended health benefits and a monthly income to patients who are unable to work. Prescription medication costs, including costs for expensive antiretroviral therapy, are not otherwise covered by a public plan and need to be paid by the individual or by their private insurance company. It is unclear to many providers and patients what happens with their drug coverage when they go back to work. This care provider explained their patients were concerned about losing drug coverage:

"I think there are a lot of people who wouldn't go and who wouldn't look into work because they're afraid of losing drug coverage and like it's just too complicated to even fathom looking at that and so universal you know universal coverage for HIV medications would certainly go a long way to helping that." [Primary care clinic, Toronto]

Another described,

"I think sometimes people are concerned about what will happen with their benefits and their coverage and sometimes they're concerned that moving to work is going to be a worse financial situation instead of a better financial situation or they simply have no idea or know who to talk to". [ID clinic, Toronto].

Another echoed this concern by saying,

"Even if they go back to work and get a drug card and it's only 80 percent [coverage] it's still a big chunk. So, that's partly why the whole drug benefit and coverage specifically for HIV because the medications are so expensive and are lifelong, is this really key part of this issue around employment." [ID clinic, Toronto]

Similarly to providers, patients expressed concerns about losing their income and their drug coverage if they decided to reenter into work. One patient said,

"It's not worth working unless it's a good paying job and then you've got to give up ODSP altogether except for the drug card because if they say well you got to pay for your own HIV meds now, now that you're working full-time that's the big thing that we haven't talked about yet. How am I going to pay for it?"[Male, Ottawa, 58].

Patients felt that drug coverage was a key issue in keeping them out of work. One said,

"At the end of the day when I look at it, it's like I might actually find myself working throughout the days a week but actually getting less than I can get right now. How could I get my drugs and how could I pay for my rent? It's sort of like keeping people in this, you know, position." [Male, Toronto, 61]

Another described,

"I see a lot of people in retail, minimum wage, you know which is slightly more than what you make on ODSP but then the benefits are the important thing you know especially if you have to pay for your HIV meds. . . you know it's like minimum wage and then if you're

paying Trillium [a governmental program which assists with high cost prescription medications] as well and your prescriptions and your dental cleaning, and you have no benefits, why go back? [Male, Toronto, 53]

**Uncertainty around provider role in assisting patients with employment.** While all providers agreed that employment was an important social determinant of health, some providers felt that assisting their patients with employment was outside the scope of their role.

"Is it my job to deal with their employment? I'm their doctor you know like could they be asked how's their, any number of things, their relationship with their children, how is their financial investments, their financial security like I just, where does my job end?" [ID clinic, Toronto].

Another explained,

"Our roles are really around the ability to assess whether or not somebody has a medical condition that prevents them from being able to be gainfully employed."[Primary Care Clinic, Toronto]

While others who were interested in assisting their patients with employment, were concerned that if they brought up this subject that they would have no way to meaningful help patients actually gain or maintain employment. This care provider said they were unsure of how to support employment needs.

"I think that's where I struggle the most is you know the patients might, I might come across their employment or not employment status based on drug coverage cause that's usually the question I ask but then when they, you know, are ready and seem interested I don't know how to kind of help bridge that or kind of lead them in the right way." [Primary care clinic, Toronto].

Providers expressed anxieties in addressing employment with patients for fear of how patients might react.

"I think the hardest thing about this discussion is actually initiating the discussion because I think that there is this perception perhaps that we are as a healthcare provider pushing people back to work and so I'm very conscious of that with my patients" [Primary care clinic, Toronto]

Another said,

"I'm guilty of that as a provider. It's sort of like you don't want to have that conversation cause it brings up you know potentially an unpleasant discussion." [Primary care clinic, Toronto].

Patients were unsure of whether their health care providers should play a role. Similarly to some providers, some of the patients suggested that they didn't have much of a role to play.

"Other than giving me a clean bill of health there's really little that the doctor can do" [Male, Toronto, 53].

Another expressed,

"I don't have to consult my doctor to go back to work; right. I don't have any physical ailments that would cause me not to get gainfully employed."[Male, Ottawa, 52].

Others thought that it would be very important to discuss with their health care team.

"Well I think talking to my doctor again would be the first step. I think he's very, he's known me for 15 years."[Male, Toronto, 54]

Another agreed that doctors played a role and said,

"What are the challenges health-wise that I present? So, they know better so I believe they can play a very significant role in that." [Female, Toronto, 50].

The willingness for patients to have their providers being involved related to the relationship they had and their trust of the healthcare system.

**Uncertainty around ability to work because of episodic disability.** A final theme that emerged from both the providers and patients was uncertainty around patients' ability to work due to the episodic nature of HIV. One provider said,

"I find that that's really complicated for people to understand why like there are some days or weeks where they just cannot work; right, and then there's other times where they might feel better or be actually more functional and more able. The challenge is often people not knowing how long that will last; right, and I think, and not being able to plan around that. . .for a lot of people the issue becomes what they're able and willing to do is not necessarily matched by what employers are willing to accommodate or willing to be flexible around which I think is often a challenge." [Primary Care Clinic, Toronto].

Similarly, patients were uncertain about working because there were periods of time where they felt unable to work. One worried about losing their job and income as a result of taking time off.

"Now they might be have a job for some days or for some week based on maybe what the doctor told them to rest and then I get to it okay, because of what you facing now, you have to rest for two weeks and that two weeks they might be out of job so what will happen to their pay?" [Female, Toronto, 42].

Another said,

"I think that's a barrier too. Yeah, cause sometimes you get, just this is something that's unpredictable. Some days you are good and some days you are bad." [Male, Toronto, 34]

A third was worried about how taking time off could be perceived by their employer.

"[I worry about] looking like you're flaky because you're not coming in like you know once a week or whatever because you're not feeling good or, and you don't want to be looked at as lazy or anything like that."[Male, Ottawa, 51]

### Potential future integration of vocational services into health services settings

Despite these uncertainties, providers expressed interest in vocational service that was co-located at their clinic. This care provider noted how the patient-health care provider relationship can be leveraged for employment supports.

"There is an existing relationship with the health provider with this building, with the people who work in it. There's a trust there. That's great. You can leverage that to increase uptake and follow through with the program." [Primary care clinic, Toronto]

However there were some uncertainties about how this type of program would be set up or organized. Providers thought that both one-on-one and group formats would be useful for this type of program. Some providers wondered whether it needs to be specific to people living with HIV.

"Why do they need a special employment service? Like it almost seems to be a conflicting message to what we're trying to tell them is that they're, they have a chronic medical illness like anybody but yet we have a special employment service." [ID clinic, Toronto]

Patients were more divided about an employment service that was located at their physician's clinic. One patient thought,

"Not a health centre for sure. . .It would just be a reminder maybe, a reminder that because I am HIV positive the service is offered in the hospital." [Male, Toronto, 46].

Others thought that an employment service linked to their physician would be valuable especially in issues around disclosing their status.

"The physician could advise HIV patients or come in and talk to HIV patients, on what not to do and what to do. For example, if accidents happen and how to communicate that issue of disclosing your status. So ya I think they have a huge role to play. Talking about the rules of when to disclose or not."[Female, Toronto, 36].

Others thought that they could advise on limitations due to their condition.

"He doesn't want me to do physical work. . .Like getting up on a ladder and washing walls because of my bone, I've also got bone density issues. I've got osteoporosis. . .I broke my wrist on some ice and it never really healed right so that is why he doesn't want me washing walls so then I would have physical limitations so he would have to do a doctor's note or recommendation. So I see him playing a role in that for sure. . . He's knows my entire history so he would know what my limitations are" [Male, Toronto, 54].

Another said,

"If it's in regards to talking about what type of job that will be less stressful for you and what they think you, advice they can give as to what you should do, what type of job what will be too risky for you and what will be strenuous on your health." [Male, Toronto, 34].

Others thought that it would be useful to have a connection between the physician and the employer and suggested that the doctors could act as advocates for their patients in relation to employment rights issues.

"Dr. X [the patient's family doctor] had to go there for the arbitration process and he told them very clearly listen, I put this restrictions for the company to accommodate him not to get him fired." [Male, Toronto, 54].

## Discussion

In this study, we describe the current and potential role of integrating vocational services into health care settings to address employment concerns for PLWH. We found that participating health care providers and patients have little experience with employment interventions beyond making or receiving referrals to community resources and providing or receiving letters for workplace accommodations to employers. This was largely due to uncertainties about continued drug coverage, the role of health providers and the episodic nature of HIV. Health care providers and some PLWH agree there is potential for a larger role for health care clinics to play in providing employment interventions for PLWH.

A main concern for both providers and PLWH was uncertainty around insurance and continued drug coverage upon returning to work. A scoping review found similar results in that PLWH on disability benefits were less likely go return to work for fear of losing benefits. Additionally, those who were on governmental assistance were often deterred by complicated processes to get back to work [43]. While this was a main concern among participants, this hesitation can be overcome with increased education and awareness as there are programs in Ontario that allow people to maintain drug benefits while re-entering into work [44,45]. However, this may not be true in other jurisdictions.

Some patients and providers were uncertain about the role of the health care team in assisting with employment. Research has found that the integrating VR services into health settings creates defined roles for each team member (including the health provider) to play to meet the patients' employment needs [38]. Integrating employment specialists into healthcare settings may address the reluctance we found in some physicians to take on added roles and responsibilities and may improve uptake of these services by PLWH. Previous research has found that successful VR programs in health settings often include communication with and buy in from employers [38]. Designing interventions to include employers could alleviate some uncertainties about repercussions of taking time off as a result of the episodic nature of HIV. This does not mean that individuals would need to disclose their status to the employer but would indicate that the employers who buy in to these programs are open to accommodations based on individual medical needs.

Our study also found that PLWH suggest that health care providers could provide advice on the disclosure of status, as well as act as advocates with employers. This finding was also present in work by the Dutch Institute for Health Care Improvement, which developed a multidisciplinary guideline that supports VR for PLWH [46]. One of the domains of this guideline is focused on disclosure and stigma. The recommendation is that health care providers provide guidance on disclosure and fear of stigmatization because PLWH tend to not disclose their status at work and nondisclosure impacts a patient psychologically [46–48]. Participants also suggested that health providers could advise on any limitations as a result of their condition. This is consistent with literature that has found that ongoing communication and sharing health needs and limits between a care provider and employment specialist as well as advising on health needs and limits is a key ingredient of successful interventions to address employment [38].

This study has strengths and limitations. A key strength is the inclusion of perspectives from both health care providers and patients. There was a variety of health care providers from different disciplines across five different clinics so a range of perspectives were captured. A

limitation of this study was that only patients who had a relationship with AIDS Service Organizations or clinical care were interviewed, therefore the perspectives of those who are less engaged in care were not captured in this study. Another limitation is that we only sampled participants from major city centres, therefore the results may not be transferrable to smaller cities or rural areas. Finally, we did not return transcripts to participants for verification and clarification nor did the participants provide feedback on the findings, therefore we are only presenting results from the initial interview or focus group.

While it is currently not the norm, there is potential for employment interventions to be integrated into health care settings. Health care providers and some PLWH agree that this could be beneficial for patients. While there were some hesitations from both patients and providers, many of these can be addressed. One immediate area for intervention is educating providers so that they can advise patients that they can maintain their drug coverage while going back to work. Other future work would be to implement and evaluate an employment intervention in health care settings for PLWH for those who were interested in receiving this type of integrated service. The design could be informed by five key features that were common in successful interventions of vocational services within health settings [38]. It would also be helpful to build off of existing VR resources for PLWH. These interventions would need to be evaluated for their acceptability and feasibility as well as their impact on quality of life, HIV outcomes and employment outcomes to understand how integrating health and vocational services for PLWH would help to address the social determinants of health.

## Supporting information

**S1 Appendix. Guide for HIV provider focus groups.**
(DOCX)

**S2 Appendix. Interview guide for PLWH.**
(DOCX)

## Author Contributions

**Conceptualization:** Amy Craig-Neil, Beth Rachlis, Claire E. Kendall, Sergio Rueda, Ann N. Burchell, Andrew D. Pinto.

**Data curation:** Amy Craig-Neil, Andrew D. Pinto.

**Formal analysis:** Amy Craig-Neil, Julia Ho, Melissa Perri, Mark Gaspar, Charlotte Hunter, Andrew D. Pinto.

**Funding acquisition:** Andrew D. Pinto.

**Investigation:** Amy Craig-Neil, Melissa Perri, Mark Gaspar, Charlotte Hunter, Andrew D. Pinto.

**Methodology:** Beth Rachlis, Claire E. Kendall, Sergio Rueda, Ann N. Burchell, Andrew D. Pinto.

**Project administration:** Amy Craig-Neil.

**Supervision:** Andrew D. Pinto.

**Writing – original draft:** Amy Craig-Neil, Julia Ho.

**Writing – review & editing:** Amy Craig-Neil, Julia Ho, Melissa Perri, Mark Gaspar, Charlotte Hunter, Beth Rachlis, Claire E. Kendall, Sergio Rueda, Ann N. Burchell, Andrew D. Pinto.

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
