## [Decision Letter · Decision Letter 0]

19 Oct 2022

PONE-D-22-23683Healthcare system action on employment as a social determinant of health in people living with HIV: A qualitative studyPLOS ONE

Dear Dr. Pinto,

Thank you for submitting your manuscript to PLOS ONE. After careful consideration, we feel that it has merit but does not fully meet PLOS ONE’s publication criteria as it currently stands. Therefore, we invite you to submit a revised version of the manuscript that addresses the points raised during the review process. Please ensure to address the reviewer comments below as well as the additional comments from the academic editor.

We look forward to receiving your revised manuscript.

Kind regards,

Emily A Hurley, M.P.H., Ph.D.

Academic Editor

PLOS ONE

Journal Requirements:

2. Please ensure that you have specified (1) whether consent was informed and (2) what type you obtained (for instance, written or verbal, and if verbal, how it was documented and witnessed). If your study included minors, state whether you obtained consent from parents or guardians. If the need for consent was waived by the ethics committee, please include this information.

4. We noted in your submission details that a portion of your manuscript may have been presented or published elsewhere. 

"The interview data for people living with HIV (PWLH) reported in this study have been used in a related manuscript published in the int. J. of Equity in Health, which focused on perceived barriers to employment from the perspective of PWLH. The manuscript we are submitting focuses on employment programs rather than barriers to employment, and includes significant additional data from health-care providers. Furthermore, the analysis for each paper was conducted separately, leading to a distinct set of themes focused on integration of health services and vocational rehabilitation. The related manuscript has been uploaded with this submission." 

Please clarify whether this publication was peer-reviewed and formally published. If this work was previously peer-reviewed and published, in the cover letter please provide the reason that this work does not constitute dual publication and should be included in the current manuscript.

Additional Editor Comments:

Thank you for this well-written manuscript. I have two comments in addition to those provided by the reviewer

- The abstract should include some information about how data was analyzed in the methods sub-section

- As reviewer 1 notes, please ensure to follow CORE-Q criteria for reporting qualitative studies and mention in limitations whenever certain CORE-Q aspects were not considered. Specifically, I would like to see some discussion of methodological orientation and theory (item no. 9) at the beginning of the methods section.

Reviewers' comments:

Reviewer's Responses to Questions

**Comments to the Author**

1. Is the manuscript technically sound, and do the data support the conclusions?

Reviewer #1: Partly

Reviewer #2: Yes

2. Has the statistical analysis been performed appropriately and rigorously? 

Reviewer #1: N/A

Reviewer #2: N/A

3. Have the authors made all data underlying the findings in their manuscript fully available?

Reviewer #1: Yes

Reviewer #2: Yes

4. Is the manuscript presented in an intelligible fashion and written in standard English?

Reviewer #1: Yes

Reviewer #2: Yes

5. Review Comments to the Author

Reviewer #1: Thank you for submitting this manuscript. While I enjoyed the content the scope of this review is to address the methodological evaluation of the submitted work. The team reported their methodology in the form of COREQ but didn't seem to consider how the aspects of recommended methodology not included in this design (e.g., not validation of findings with participants) can or should impact the study's conclusions. This lack of integration is a bit of a disconnect and once that could be addressed in the discussion. Further, identifying pts and providers different (i.e., city or clinic, some w/ age) is also inconsistent. I have no additional comments on the work's methodology.

Reviewer #2: Thank you for the opportunity to review this excellent manuscript. I recommend publication with very minor suggestions for improvement.

• In the introduction, you note “However there are many barriers including88 uncertainty around losing disability benefits, particularly coverage for expensive antiretroviral89 therapy, and a fear of being trapped in a job to then maintain drug benefits (19,22–24).” Consider noting that these concerns vary by country, state/province. In the discussion you might want to come back to these concerns to comment on the fact that these drug coverage policies creating barriers to employment are political and can be changed, i.e., when you explain that in Ontario these concerns are unfounded (even though most people are unaware of this fact) you can use this as an opportunity to emphasize that in a context without ongoing pharmaceutical coverage for PLWH even once they are employed, employment assistance services really are not appropriate or feasible. Basically, Ontario has created the enabling policy environment so it can proceed to the next steps of raising awareness about this positive policy and then helping people actually access and navigate employment. But in the US, in places where coverage really would end if you got a job, you could not proceed with employment assistance.

• In the methods, a very brief description of the domains covered by the interview and FGD guide would be useful. I know you uploaded the guides as supplementary files but an in text summary would be helpful.

• In table 1, please check the %. I.e. 2 participants in Ottawa were 18-25 years old… 2/20 should be 10% rather than 0%.

• I think one area that could be highlighted in the discussion is that doctors have a role to play in explaining to patients that they can maintain their pharmaceutical coverage while going back to work. Doctors themselves were unaware of this program, so that strikes me as a key area for immediate intervention.

• You suggest that “Designing interventions to include employers could alleviate some452 uncertainties about repercussions of taking time off as a result of the episodic nature of HIV.” It may be worth additional reflection on how this involvement of employers can or should be done in a manner that avoids disclosing employee HIV status or increasing stigma. Perhaps you could draw from literature on employment accommodation for people with other chronic conditions.

• You note “Our study also found that PLWH suggest that health care providers should provide advice on the455 disclosure of status, as well as act as advocates with employers” – but was this a widespread sentiment? You may want to say that some PLHIV felt this way, but it seems likely that others did not want their doctor involved in their employment because they did not want emphasis placed on their HIV status. Is it a bit naïve to think that all employers will respond well to learning that their employee is living with HIV and that disclosure at the workplace is good for all employees?

• You note “While there were some hesitations from both patients and providers,478 many of these can be addressed.” You could further note that employment interventions integrated into healthcare settings would not be for all PLWH and thus could be there for those who want them and ignored by those who do not, rather than suggesting that with the right design these interventions could appeal to all PLWH.

6. PLOS authors have the option to publish the peer review history of their article (what does this mean?). If published, this will include your full peer review and any attached files.

Reviewer #1: No

Reviewer #2: No

---

## [Author Response · Author response to Decision Letter 0]

2 Dec 2022

The attached file, Response to Reviewers, contains a point-by-point response to the specific Editor and Reviewer comments in table format. Below we have included text of our letter and point-by-point responses, however we would suggest referring to the attached table to more clearly understand the changes.

--

Dear PLoS ONE Editorial Team, 

Thank you for taking the time to review our manuscript entitled, "Healthcare system action on employment as a social determinant of health in people living with HIV: a qualitative study", and provide us with feedback. We have addressed all editor and reviewer comments in the revised manuscript and think the paper is stronger as a result. Please see a detailed response to the comments below.

We look forward to hearing from you about this manuscript.

--

EDITOR - Journal Requirements:

#1 - Re: formatting requirements

Response: We have updated the manuscript based on the PLOS ONE style requirements.

#2: Re: Consent

Response: The following has been added, “Written informed consent was obtained for in-person interviews and focus groups. Verbal informed consent was obtained using a verbal consent checklist completed by the facilitator for virtual interviews.” [p. 8]

#3: Re: Funding information/financial disclosure

Response: The Funding Information section only allows us to select one funder and input the number however in our financial disclosure we have put all co-author full funding statements. 

#4: Re: Portion presented elsewhere

Response: The document that was uploaded with the submission was peer-reviewed and published in the International Journal for Equity in Health. As we noted in our cover letter, we do not believe that this constitutes dual publication because the focus of each paper is very different. The other manuscript focused only on barriers to employment from the perspective of PLWH. This manuscript focuses on employment programs and includes additional data from health care providers. Furthermore, the analysis for each paper was conducted separately, leading to a distinct set of themes focused on integration of health services and vocational rehabilitation. 

Re: Reference List

Response: We have reviewed the reference list. No cited papers have been retracted. Minor changes have been made to ensure consistency with the journal style. [pp. 24-27]

EDITOR- General

Re: Abstract

Response: Thank you for your comment. This information has been added. [p. 2]

Re: COREQ

Response: Thank you for your comment. This information has been added. [p. 7]

REVIEWER #1

Comment 1/1 (Re: Methodology)

Thank you for this comment. We have added a limitation to the discussion noting that we did not return transcripts to participants nor validate findings with them. 

We did not identify specific providers in the transcripts of the focus groups therefore cannot present information on gender or age in the quotations. We believe that there is value to have this information remain in the manuscript despite the fact that it is different between the two groups. It is however consistent within the groups (of providers and patients). [p. 22]

REVIEWER #2:

Comment 1/7 (Re: Jurisdictional variation)

Thank you for this comment. We have updated the manuscript accordingly to note that this policy is specific to Ontario and may not apply to other jurisdictions. [p. 4 & 21]

Comment 2/7 (Re: Interview guides)

Response: Thank you for this comment. This has been added. [p.8]

Comment 3/7 (Re: Table 1)

Thank you for this comment. This has been corrected. [p. 10]

Comment 4/7 (Re: MDs awareness of program)

Thank you for this comment. We have added this to the discussion. [p. 22]

Comment 5/7 (Re: Employers & disclosure)

Response: Thank you for this comment. We have added an additional reflection on avoiding forced disclosure when employers are involved. 

[p. 21]

Comment 6/7: (Re: Disclosure of status)

Thank you for this comment. We have adjusted this sentence to say that health care providers "could" (rather than "should") provide advice. 

Comment 7/7 (Re: Non-applicability to some PWLH)

Thank you for this comment. We have added this consideration to the manuscript. [p. 22]

---

## [Decision Letter · Decision Letter 1]

15 Feb 2023

Healthcare system action on employment as a social determinant of health in people living with HIV: A qualitative study

PONE-D-22-23683R1

Dear Dr. Pinto,

We’re pleased to inform you that your manuscript has been judged scientifically suitable for publication and will be formally accepted for publication once it meets all outstanding technical requirements.

Kind regards,

Emily A Hurley, M.P.H., Ph.D.

Academic Editor

PLOS ONE

Additional Editor Comments (optional):

Reviewers' comments:

Reviewer's Responses to Questions

**Comments to the Author**

1. If the authors have adequately addressed your comments raised in a previous round of review and you feel that this manuscript is now acceptable for publication, you may indicate that here to bypass the “Comments to the Author” section, enter your conflict of interest statement in the “Confidential to Editor” section, and submit your "Accept" recommendation.

Reviewer #2: All comments have been addressed

2. Is the manuscript technically sound, and do the data support the conclusions?

Reviewer #2: Yes

3. Has the statistical analysis been performed appropriately and rigorously? 

Reviewer #2: N/A

4. Have the authors made all data underlying the findings in their manuscript fully available?

Reviewer #2: Yes

5. Is the manuscript presented in an intelligible fashion and written in standard English?

Reviewer #2: Yes

6. Review Comments to the Author

Reviewer #2: The authors have sufficiently addressed my comments. I am satisfied with their response.

Best wishes.

7. PLOS authors have the option to publish the peer review history of their article (what does this mean?). If published, this will include your full peer review and any attached files.

Reviewer #2: No

---

## [Editor Report · Acceptance letter]

30 Mar 2023

PONE-D-22-23683R1 

Healthcare system action on employment as a social determinant of health in people living with HIV: A qualitative study 

Dear Dr. Pinto:

I'm pleased to inform you that your manuscript has been deemed suitable for publication in PLOS ONE. Congratulations! Your manuscript is now with our production department. 

Kind regards, 

on behalf of

Dr. Emily A Hurley 

Academic Editor

PLOS ONE